# Effect of Temperature, Surface, and Medium Qualities on the Biofilm Formation of *Listeria monocytogenes* and Their Influencing Effects on the Antibacterial, Biofilm-Inhibitory, and Biofilm-Degrading Activities of Essential Oils

**DOI:** 10.3390/foods14122097

**Published:** 2025-06-14

**Authors:** Anita Seres-Steinbach, Péter Szabó, Krisztián Bányai, György Schneider

**Affiliations:** 1Department of Medical Microbiology and Immunology, Medical School, University of Pécs, H-7624 Pécs, Hungary; seres-steinbach.anita@edu.pte.hu; 2Department of Geology and Meteorology, Faculty of Sciences, University of Pécs, Ifjúság str. 6, H-7624 Pécs, Hungary; sz.piiit01@gmail.com; 3Department of Pharmacology and Toxicology, University of Veterinary Medicine, H-1078 Budapest, Hungary; bkrota@hotmail.com; 4National Laboratory of Infectious Animal Diseases, Antimicrobial Resistance, Veterinary Public Health and Food Chain Safety, University of Veterinary Medicine, H-1078 Budapest, Hungary; 5Department of Medical Biology, Medical School, University of Pécs, H-7624 Pécs, Hungary

**Keywords:** *Listeria monocytogenes*, biofilm eradication, essential oils, inhibitory, antibacterial effect, temperature dependence

## Abstract

*Listeria monocytogenes* is a foodborne pathogen with a high tolerance to a wide range of environmental conditions, making its control in the food chain a particular challenge. Essential oils have recently been considered as potential antilisterial agents. In this study, the antilisterial effects of 57 EOs were tested on 13 different *L. monocytogenes*. Thirty-seven EOs were found to be effective in a strain and temperature-dependent manner. At 37 °C, all EOs were effective against at least one strain of *L. monocytogenes*. However, at 14 °C and 23 °C, 12 EOs, such as Minth, Nutmeg, Neroli, Pepperminth, etc., became drastically ineffective. The efficacy of the EOs increased at the lowest temperature, as only four EOs, such as Dill seed, Juniper, lemon eucalyptus, and sandalwood, were found to be ineffective at 4 °C. Ajowan and thyme were the only EOs that were antibacterial against each strain at all temperatures tested (4, 14, 23, 37 °C). Biofilm-inhibition tests with 57 EOs, performed on polystyrene plates with different surface qualities and stainless steel, using 0.1% and 0.5% final concentrations, showed the outstanding inhibitory abilities of ajowan, geranium, Lime oil, melissa, palmarosa, rose geranium, sandalwood, and thyme. Fennel, lemon eucalyptus, and chamomile had the potential to inhibit biofilm formation without affecting live bacterial cell counts. Ajowan, geranium, thyme, and palmarosa reduced the biofilm to the optical density of 0.0–0.08, OD: 0.0–0.075, 0.0–0.072, and 0.0–0.04, respectively, compared to the bacterium control 0.085–0.45. The mature antibiofilm eradication ability of the EOs revealed the outstanding features of ajowan, geranium Lime, melissa, palmarosa, rose geranium, and thyme by suppressing the established biofilm to one tenth. The different sensitivities of the isolates and the temperature-dependent antilisterial effect of the tested EOs have to be taken into account if an EO-based food preservation technology is to be implemented, as several *L. monocytogenes* become resistant to different EOs at medium temperature ranges such as 14 °C and 23 °C.

## 1. Introduction

*Listeria monocytogenes* is a ubiquitous bacterium that can be isolated from different environments, such as soil [1], natural waters, and wastewater [2]. This bacterium has the ability to colonize the intestinal system of animals and humans [3]. It has been isolated from chicken [4], pig [5], and cattle flocks [6], wild animals such as red deer and wild boar [7], rodents [8], fish [9,10], and also from crops, fruit, and vegetables [11,12]. In humans, colonization can be asymptomatic and last for a long time. In certain cases, however, symptoms occur during infection, which is called listeriosis. Non-invasive forms of listeriosis are confined to the gastrointestinal tract and cause milder symptoms [13], while in some cases, *L. monocytogenes* can cross the intestinal barrier (such as the lamina propria) and spread to the bloodstream, where it can cause sepsis, and by reaching the blood–brain barrier, it can cause the life-threatening condition of meningitis [14]. During pregnancy, the bacterium is able to cross the feto-placental barrier, and can cause birth defects or miscarriage [15].

Listeriosis is typically associated with the consumption of contaminated foods, such as cheese, meat, vegetables, and fruit [16,17]. Its widespread occurrence, virulence potential, and wide tolerance for low (−0.4 °C) and high (+50 °C) temperatures [18], acidic (pH: 4), alkaline (pH: 9) [19], and saline conditions (up to 20%), make *L. monocytogenes* one of the most important microbiological hazards in the food industry. It is estimated that in the USA alone, the annual economic burden of foodborne listeriosis is around USD 4 billion [20] and around EUR 2.4 billion [21], due to medical costs, productivity losses, and the loss of consumer confidence. This is the reason why in several countries, there is a zero tolerance policy against this bacterium species in meat products [22,23], and this is the reason why several recent studies have focused on the elimination of this bacterium from different foods, such as fresh and chilled meats, salads, ice creams, fruits, and juices [24,25,26].

An important aspect of *L. monocytogenes* survival is its ability to form biofilms on either biotic or abiotic surfaces [27,28]. This ability of the bacteria is influenced by several environmental factors, such as temperature, pH, salinity, nutrient availability [29,30], and surface qualities such as glass, stainless steel, and polystyrene [31]. Biofilm formation is a major challenge in the food processing sector, as biofilm protects bacteria from external stresses, such as nutrient deprivation, and renders the resident bacteria resistant to various disinfectants. Through this strategy, the bacterium has an increased ability to survive on meat surfaces or slaughterhouse equipment, making cross-contamination possible [32,33,34]. Therefore, elimination practices must target not only the bacteria themselves, but also the complex matrix of the biofilm itself, for which the use of ultrasound, ozone [35,36], and bacteriophages [37] have recently been proposed. The use of essential oils, due to their antimicrobial properties and defining flavor characteristics, also offer a practical and potential approach for the elimination of *L. monocytogenes* cells from various surfaces.

The efficacy of certain essential oils has been tested on *L. monocytogenes* in vitro [38,39,40], and also on different food matrices [41,42,43,44], but the applied foods, essential oils (EOs), concentrations of EOs, temperatures, and treatment methods used were so diverse that the results are not so easily comparable, as individual strains with varying biofilm-forming capacities and different sensitivities to EOs were used for these and other previous studies. Several studies were carried out at only one temperature, despite the fact that this zoonotic bacterial species is not only mesophilic, but also psychrophilic and therefore able to survive and multiply at different temperatures.

Nevertheless, recent experiments focusing on the inhibition of biofilm formation were also carried out in different media, most frequently in Brain Heart Infusion and Mueller Hinton II broth [45,46,47].

In order to focus on the potential applicability of essential oils and to obtain a clear picture of their efficacy under different circumstances, we performed a didactic study with 13 different *L. monocytogenes* isolates and 57 EOs. The experiments compared the growth kinetics of the isolates in different laboratory media (Mueller Hinton, Luria Bertani, and Brain Heart Infusion) at different temperatures (23 °C, 37 °C), while their biofilm-forming capacities were compared in different laboratory media (Mueller Hinton, Luria Bertani, and Brain Heart Infusion) and chicken meat juice at different temperatures (4 °C, 14 °C, 23 °C, 37 °C), and on polystyrene surfaces with different adhesive properties and on stainless-steel mesh. The antibacterial, biofilm-inhibitory, and eradication effects of the EOs were investigated at 4 °C, 23 °C, and 37 °C on the above-mentioned surfaces. These experiments were performed with essential oils at final concentrations of 0.1 and 0.5%. Furthermore, the effects of the most promising essential oils on morphology were investigated by SEM. Our results provide an insight into the potential of different EOs and the effects that influence their efficacy, which could contribute to the development of innovative EO-based strategies to ensure food safety at different stages of food processing and transport.

## 2. Materials and Methods

### 2.1. Bacterial Strains, Media

Thirteen different *L. monocytogenes* isolates, one reference strain (ATCC 35152), and twelve isolates from different sources were used in the experiments (Table 1). The strains were selected from our strain collection, to contain reference strains and local isolates from foods. The bacterial strains were routinely grown on Luria Bertani (LB) agar plates. Prior to experiments, individual clones were inoculated into 5 mL of LB broth and incubated for 18 h at 37 °C in a shaking thermostat (120× rpm). Bacterial cell densities were synchronized to OD620 = 0.2 (approximately 108 CFU/mL) using a spectrophotometer. The resulting bacterial suspensions were used for the downstream processes described below. Müller Hinton II (MHII, Biolab Co., Budapest, Hungary) and Brain Heart Infusion (BHI, Biolab Co., Hungary) liquid media were used for growth kinetics and biofilm-formation experiments. Chicken meat juice (CMJ) was used for some experiments. CMJ was prepared from fresh chicken breast before the experiments. To 100 g of chicken meat, 200 mL of 0.9% NaCl solution was added and minced. After centrifugation (10,000× *g*, 10 min), the supernatant was sterile filtered, and the meat juice obtained was used in further experiments.

### 2.2. Essential Oils

Fifty-seven essential oils were purchased from A.G Industries (Noida, UP, India) for the tests, such as ajowan oil (*Trachyspermum ammi*), Anethole oil (*Foencukum vulgare mill*), basil oil (*Ocimum basalicum*), Bay oil (Laurus nobilis), Black pepper oil (*Piper nigrum*), calamus oil (*Acorus calamus*), chamomilla (*Matricaria chamomilla*), Cajeput oil (*Melaleuca leucadendron*), Cedarwood oil (*Juniperus ashei*), Cardamom oil (*Elettaria cardamomum*), cinnamon oil (*Cinnamomum zeylanicum*), cinnamon leaf oil (*Cinnamomum zeylanicum*), Citronella oil (*Cymbopogon nardus*), Clary Sage oil (*Salvia sclarea*), Cypress oil (*Cupressus sempervirens*), Dill seed oil (Anethum graveolens), eucalyptus oil (*Eucalyptus globulus*), fennel oil (*Foeniculum vulgare*), fenugreek oil (*Trigonella foenum*-graecum), frankincense oil (*Boswellia serrata*), geranium oil (*Pelargonium graveolens*), Ginger oil (*Apium graveolens*), Grapefruit oil (*Citrus Paradisi*), Jasmine oil (*Jasminum gradiflora*), Juniper oil (*Juniperus communis*), Bulgarian lavender oil (*Lavandula angustifolia*), lemon oil (*Citrus limanum*), lemon eucalyptus oil (*Eucalyptus citriodora*), Himalayan lavender oil (*Lavandula angustifolia*), Lime oil (*Citrus aurantifolia*), Lime oil as per BP (*Citrus Aurantifolia*), melissa oil (*Melissa officinalis*), Minth oil (*Mentha spicata*), Myrtle oil (*Myrtus communis*), Neroli oil (*Citrus aurantium*), Nutmeg oil (*Myrstica fragans*), Orange (*Citrus sinensis*), palmarosa oil (*Cymbopogon martini*), patchouli oil (*Pogostemon cablin*), Pepperminth oil (*Mentha piperita*), petitgrain oil (*Citrus aurantium*), pine oil (*Pinus sylvestris*), Ravensara oil (*Ravensara aromatica*), rose geranium oil (*Rosmarinus officinalis*), Saffron oil (*Crocus sativus*), sandalwood oil (*Santalum album*), Sage oil (*Salvia officinalis*), tarragon oil (*Artemisia dracunculus*), tea tree oil (*Melaleuca alternifolia*), Thuja oil (*Thuja occidentalis L.*), Tolu Balsam (*Myroxylon balsamum*), Turmeric oil (*Curcuma longa*), Wintergreen oil (*Gaultheria fragrantissima*), Ylang Ylang oil (*Cananga odorata* var *genuine*), Vanilla (*Vanilla planifolia*), Vetiver oil (*Vetiveria zizanoides*), Mace oil (*Myristica fragrans*), and thyme (*Thymus vulgaris*). The batches of all EOs were quality certified. EOs were used either in concentrated or diluted form, as indicated in the relevant sections. One and five percent emulsions were prepared from the concentrated essential oils in 1% Tween 20 solution used for testing. The EO suspensions were regularly tested for sterility.

### 2.3. Drop-Plate Testing

For antibacterial screening, 100 µL of OD620 = 0.2 bacterial suspension was spread on the surface of Columbia Blood Agar base plates using a sterile triangular glass rod. After drying, 5 µL (1%) of the essential oils were dropped on the bacterial lawns and incubated at 4 °C, 14 °C, 23 °C, and 37 °C. The size of the inhibition zones was measured after 24 h in the plates incubated at 23 °C and 37 °C, after 48 h in the plates incubated at 14 °C, and after four days in the plates incubated at 4 °C [48].

### 2.4. Comparative Growth Kinetics of L. monocytogenes Isolates

Comparative growth kinetics experiments with the 13 *L. monocytogenes* isolates were performed in non-adhesive 96-well tissue culture plates (83.3924.500, Sarstedt, Nümbrecht, Germany). Three different liquid media, LB, MH-II, and BHI, and two different temperatures, 23 °C and 37 °C, were used for the experiments. At the beginning, 198 µL of media were pipetted into the respective wells, to which 2 µL of the OD620 = 0.2 bacterial suspensions were added and suspended. The plate thus prepared for the experiment was placed in a temperature-controlled multimode reader (Allsheng, Hangzhou, China) for 24 h and changes in optical densities at a wavelength of 630 nm were monitored every 15 min. A shaking step of 5 s was incorporated before each measurement. The data obtained were converted into Excel files and the curves were plotted on graphs. Monitoring under all conditions was performed in triplicate (3 wells/isolates). Our experiments were carried out with all isolates, as this is the only way to obtain a more accurate picture to draw more general conclusions about the effect of medium and temperature on isolates.

### 2.5. Biofilm-Formation Tests

The biofilm-formation capacities of the 13 *L. monocytogenes* isolates were determined at 4 °C, 23 °C, and 37 °C on 96-well polystyrene cell culture plates (Sarstedt, Nümbrecht, Germany) with 3 different adhesion properties, such as adhesive (plate code: 83.3924), complex adhesion cells (plate code: 83.3924.300), and suspension cells (plate code: 83.3924.500), abbreviated as A, CAC, and SC, respectively, and on stainless-steel mesh (SSM) plates (0.27 × 0.27 × 0.16; Metmark Kft, Szekszárd, Hungary) of a uniform size (25 mm^2^). Tests were performed in different media, such as LB, MH-II, and BHI, and chicken meat juice (CMJ), and the applied EO concentration was 0.1 and 0.5%.

For the tests, overnight (ON) bacterial suspensions (OD620 = 0.2) were diluted 100 times in LB, MH-II, or BHI medium. Two hundred microliters of the dilutions were added to the wells of 96-well plates and the plates were incubated at 23 °C and 37 °C for 18 h and at 4 °C for 3 days. After incubation, non-adherent cells were carefully removed, and the wells were carefully washed 3 times with PBS. Sessile cells were fixed with 200 μL 2% formalin (28.6 mL 35% formalin in 471.4 mL PBS) for 2 min. After the removal of formalin, the plates were dried at 37 °C for 2 h. Fixed cells were stained with 0.13% crystal violet (14.29 mL 35% formalin + 234.41 mL PBS + 1.302 mL 96% ethanol + 0.325 g crystal violet) for 20 min and then washed three times with PBS. Fixed and crystal violet-stained cells were solubilized with 200 µL of 1% SDS (50% ethanol + 50% PBS) for 2 h, and optical density values were read at 630 nm using a multimode reader (AMR-100T, Hangzhou, China).

To investigate the biofilm-forming capacity of the isolates on SSM, the plates were disinfected with 96% ethanol for 30 min and then autoclaved. The plates were placed in the wells of 24-well tissue culture plates and incubated as described above. After incubation, the SSMs were washed 3 times in PBS and fixed in 2% formalin as described above. Fixed SSMs were examined by scanning electron microscopy [49].

### 2.6. Scanning Electron Microscope (SEM) Analysis

To visualize adherent bacteria or the formed biofilm on the surfaces of SSS, SEM analysis was performed as previously described [50]. Briefly, air-dried samples were coated with gold by using a Jeol JFC-1300 auto fine coater (Jeol, Tokyo, Japan) and samples were visualized using a Jeol JSM-IT500HR (Jeol, Tokyo, Japan) SEM in the secondary electron mode. Images were taken at different magnifications using an accelerating voltage of 5 kV and a probe current of 45 kV [51].

### 2.7. Biofilm-Inhibition Capacity of EOs

Biofilm-inhibition assays were performed on two strains, including 1860 and ATCC 35152. The method described above (2.5) was carried out with modifications. Essential oils were added to the bacterial suspensions at the beginning of the incubation at final concentrations of 0.5% and 0.1%. The tests were carried out using CACs at 23 °C and 37 °C for 1 day and at 4 °C for 3 days, and then the live bacterial counts were determined by serial dilutions, while crystal violet staining was performed on parallel wells.

### 2.8. Mature Biofilm Eradication Capacity of EOs

The biofilm-disrupting capacity of the essential oils was determined in 96 plates on mature biofilms. The procedure was very similar to the biofilm test described above (2.5), but after the appropriate incubation times (1 day at 23 °C and 37 °C, and 3 days at 4 °C), the medium was removed from the established biofilms and then carefully washed three times with PBS. Next, 200 µL of the respective sterile liquid media (MH-II) and the tested essential oils were added at final concentrations of 0.1% and 0.5%. After one day of incubation, the biofilm fixation and staining procedure described above was carried out, and the amount of crystal violet stain retained and dissolved was determined optically. After the biofilm removal test, 96-well plates containing 0.5% essential oils and performed in MH-II were sampled, the CFU of biofilms were determined, and bactericidal effects at all 3 temperatures were revealed [49].

### 2.9. Statistical Analysis

Statistical analyses were performed using JASP 0.18.3.0, where we sought to demonstrate the relationship between the effects using linear regression, with 95% confidence intervals. The relationship between temperature and optical density was constructed by plotting marginal effects. Comparisons between temperatures, medias, isolates, essential oils, concentrations of essential oils, and surfaces were made using ANOVA. The most commonly used significance threshold for our analyses was *p* < 0.05. Factors influencing biofilm formation were analysed using IBM SPSS Statistics v.26.

### 2.10. Flow Chart of the Experimental Design

Due to the complexity of this study, here, we present the flowchart (Figure 1) of our work, outlining the experimental design. For our study, 13 *L. monocytegens* strains were isolated from different sources and partially characterized by determining their growth kinetics, biofilm-forming capacities in different media, and different temperatures. The antibacterial effect of fifty-seven concentrated EOs were performed on one *L. monocytogenes* strain, and only the effective EOs were further investigated at different temperatures at the end concentration of 1%. Based on the obtained data, the biofilm degradation potentials of the 37 EOs were investigated at different temperatures. Data were compared and statistically analyzed.

## 3. Results

### 3.1. Drop-Plate Testing

Prescreening the concentrated (100%) EOs at 4 °C, 14 °C, 23 °C, and 37 °C on the lawn of the reference *L. monocytogenes* strain 2262 (ATCC 19111) has revealed that 20 EOs were completely ineffective against this isolate. These 20 EOs were taken out from further analysis.

Of the 57 (1% concentration) essential oils tested, 20 proved to be completely ineffective at the four different temperatures applied (4 °C, 14 °C, 23 °C, and 37 °C). Ajowan, cinnamon, and thyme were antibacterial at all temperatures when applied at 1% concentration, while the antibacterial efficacy of the remaining 37 EOs showed a strain and temperature-dependent feature (Figure 2).

At 37 °C, 17 EOs were effective against at least 10 of the 13 *L. monocytogenes* isolates (Figure 2). This number was nine, six, and nineteen at 23 °C, 14 °C, and 4 °C. At 37 °C, all EOs were effective against at least one *L. monocytogenes* strain. However, 12 EOs were completely ineffective at 23 °C and the same number was determined at 14 °C. The efficacy of the EOs increased at the lowest temperature, since at 4 °C, only four EOs were found to have no antibiotic activity against any of the *L. monocytogenes* isolates.

At 37 °C and 4 °C, seventeen and nineteen of the thirty-seven EOs were effective against at least ten *L. monocytogenes* isolates, while at 23 °C and 14 °C, this number was nine and six, respectively.

In general, 37 °C and 4 °C were the most restrictive temperatures, as these temperatures supported the highest proportion of antilisterial effects of the EOs tested. From this point of view, 4 °C was more supportive, as the zones of inhibition were larger, and the effects were more balanced.

It can be clearly seen that the antilisterial effect of EOs is much more pronounced at low and high temperatures, such as 4 °C and 37 °C (Figure 2), since more isolates were resistant to the tested essential oils and there was an isolate-dependent sensitivity, except for the above-mentioned EOs such as ajowan, cinnamon, and thyme. In general, the reference strain (ATCC35152) was the most sensitive to essential oils.

### 3.2. Growth Kinetics

Comparative analyses of the growth characteristics of 13 *L. monocytogenes* isolates showed that the medium quality was the main determinant of the growth characteristics of the isolates, rather than temperature, although differences between isolates could be detectable (Figure 3).

Using LB medium, the optical densities approached OD 0.2 after 15 h of incubation. MH-II supported the growth of the isolates slightly better, helping them to reach and exceed an OD of 0.3. A more pronounced supportive effect was observed when cells were grown in BHI medium at either 23 °C or 37 °C. At these temperatures, the optical density values of the isolates were between OD 0.37 and 0.74.

Differences in growth between strains could also be observed, as the curves of the 13 different *L. monocytogenes* isolates showed more variation when isolates were grown in MH-II and BHI compared to LB medium.

### 3.3. Biofilm-Forming Capacity of the L. monocytogenes Isolates on Polystyrene Surfaces

The biofilm-formation capacities of the 13 *L. monocytogenes* isolates were compared at different temperatures, such as 4 °C, 14 °C, 23 °C, and 37 °C, in different media, such as LB, MH-II, BHI, and CMJ. All experiments were performed in three different 96-well tissue culture plates with different adhesive properties.

Irrespective of the growth medium used, it was found that the LB medium did not convincingly support biofilm formation at 4 °C, 14 °C, and 23 °C (Figure 4(A1,A2,A3)), but this changed drastically at 37 °C, which was supportive irrespective of the tissue culture plates used (Figure 4(A4)). In contrast, MH-II had a more convincing supportive effect on biofilm formation, although this effect was also most pronounced at 37 °C and significant differences between isolates could be revealed. The characteristic effects of the applied surface were observed (Figure 4(B2,B4)). In the case of BHI, characteristic differences in biofilm formation were observed on different surface qualities at 4 °C and 14 °C (Figure 4(C1,C2)), as biofilm formation was more pronounced on the stainless-steel and the non-adhesive plates. CMJ proved to be ineffective in supporting biofilm formation at any of the low and medium temperatures (Figure 4(D1,D2,D3)), but became highly supportive at 37 °C (Figure 4(D4)).

In terms of plate quality, strains 1822 and 1830 formed the best biofilm on CAC and the least on SC cells. Strains 1834, 1835, 1966, and 1994 formed biofilms equally well on SC and A cells, while their biofilm formation was worse on CAC cells. Isolates 4a, 2262, ATCC 35152, and 1780 formed the best biofilm on SC cells. It is also important to mention that a stronger biofilm activity was observed for all strains on the cell culture plates used for the cultivation of SC and CAC cells. In general, it can be said that SSS supported biofilm formation, but this property was highly dependent on the medium used, the temperature, and also the isolate (Figure 4 and Figure 5). In this respect, the best supporting temperature for SSS was 14 °C, irrespective of the medium tested.

In general, the most intensive biofilm formation was observed at 37 °C (Figure 4 and Figure 5). At this temperature, there were no drastic differences between the LB, MH-II, BHI, and CMJ used (Figure 4(A4–D4)). In contrast to 37 °C, clear differences in biofilm formation were observed at low and medium temperatures (4 °C, 14 °C, and 23 °C) (Figure 4(A1–D3)).

Individual differences between isolates were also observed. The best example of this was isolate 1934, which showed a generally low ability to form biofilms irrespective of the nutrient fluid and temperature, whereas isolates 1994 and 2262 formed a massive biomatrix at least at medium and high temperatures such as 23 °C and 37 °C (Figure 4(A3–D3) ). One isolate (1860) had the ability to form biofilm at lower temperatures (Figure 4(A1–D2)), and this ability was also observed in the LB medium.

The biofilm-formation ability of *L.monocytogenes* 1860 and 35152 used as a reference was detected on stainless steel using SEM.

At 4 °C, the reference isolate does not form biofilms in the media used. Some segregated colonies are shown in the pictures (Figure 5A–C). In the case of 1860, no biofilm is visible here either, but in BHI, clustering is already visible at the top of the image (Figure 5M–O). These results are in agreement with Figure 4(A1–D1), where the optical densities were 0.02 or less.

At 14 °C, we already see interconnected cells, e.g., in LB for the reference strain (Figure 5D–F), while in the 1860 strain, we see even more pronounced connections between bacterial cells. At this temperature, not only segregated, separate, 1-1 bacterial cells can be seen (Figure 5P–R). Optical density also showed higher values with the crystal violet method: OD: 0.02–0.08 (Figure 4(A2–D2)).

At 23 °C, again, in the 35152 strain, we see only isolated 1-1 bacterial cells in LB and BHI, but in MH-II, we see a denser, more connected multitude of bacterial cells (Figure 5G–I). Looking at the crystallographic results, similar to the SEM image, MH-II 35152 showed the highest optical density of 0.218. The biofilm formed by 1860 did not reach an OD of 0.1 in either case.

At 37 °C, 35152 did not form significant biofilms in any of the media. The images show 10–20 bacterial cells (Figure 5J–L). Strain 1860 shows more cells in MH-II, while LB and BHI show a multiplicity of bacterial cells (Figure 5V–X). When treated with crystal violet, the optical density ranged from 0.051 to 0.283 for 35152, while for 1860 it was from 0.079 to 0.219 (Figure 4(A4–C4)).

A Statistical analysis revealed a linear regression between biofilm formation and temperature. The higher the temperature, the more biofilm was formed, so increasing the temperature in the range studied (4–37 °C) also increased biofilm formation (Figure 6).

However, not only the temperature but also the type of carrier broth used had an influencing role on biofilm formation. Of the broths used, MH-II was the most supportive of biofilm formation, while CMJ and LB were less supportive, although this was influenced by temperature (Figure 7).

The analysis of variance revealed significant differences between the results of strains, broths, and temperatures in terms of biofilm formation, which was also supported by the ANOVA (*p* < 0.05). The statistical analysis confirmed that biofilm formation is a multifactorial process influenced by temperature, carrier medium, isolate used, and last but not least, the surface properties of the disc.

In SPSS, using K-Means Cluster Analyses, we made four clusters of temperature, media, plate, and optical density values, and found that BHI, non-adhesive plates, and 37 °C (0.2365) were the most conducive to biofilm formation. Of these external factors, media had the greatest effect on biofilm formation (BHI: 0.76), followed by temperature (37 °C: 0.24). Plate type had the least effect on biofilm formation, with no significant difference between plate types (*p* = 0.848).

### 3.4. Biofilm-Inhibition Assay

A concentration of 0.5% of ajowan, Bay, chamomilla, cinnamon (bark, leaf), geranium, Lime, melissa, Myrtle, palmarosa, patchouli, Peppermint, Neroli, rose geranium, sandalwood, and thyme essential oils at 37 °C not only inhibited biofilm formation of isolate 1860, but it also had a bactericidal effect in MH-II and BHI. However, some essential oils were able to inhibit biofilm formation at concentrations as low as 0.1%: ajowan, cinnamon leaf oil, geranium, Lime oil as per BP, melissa, Orange, palmarosa, rose geranium, sandalwood, and finally thyme. (Figure 8A). The more effective the essential oil was in reducing CFU, the lower the biofilm-formation efficiency. The initial bacterial count (9 × 10^8^ CFU/mL) was reduced to 10^2^ by fenugreek and Himalayan lavender oil, while frankincense, pine, lemon, Myrtle, Neroli, Ravensara, and Thuja reduced the initial CFU with 50% efficiency. In BHI at 37 °C, ajowan, Bay leaf, cinnamon leaf, geranium, Lime oil as BP, melissa, palmarosa, rose geranium, and thyme reduced the CFU to 0. Initial bacterial counts (9 × 10^8^ CFU/mL) were reduced to 10^2^ by sandalwood and to 10^4^ by patchouli. When the final concentration of essential oils was reduced to 0.1%, ajowan, cinnamon (bark, leaf), geranium, melissa, palmarosa, sandalwood, and thyme completely inhibited biofilm formation, while chamomilla reduced the CFU to 10^2^, rose geranium reduced the CFU to 10^3^, and Lime oil according to BP and Saffron reduced the initial CFU from 9 × 10^8^ CFU/mL to 10^4^/mL in MH-II at 37 °C. In BHI at 37 °C, ajowan, Black pepper, cinnamon bark and leaf, geranium, and thyme reduced the CFU to 0 and thus inhibited biofilm formation. At 23 °C and 4 °C, fenugreek reduced the initial bacterial count from a minimum (9 × 10^8^ CFU/mL) to 10^8^.

A concentration of 0.5% of ajowan, Bay, cinnamon (bark, leaf), geranium, melissa, Minth, palmarosa, patchouli, rose geranium, sandalwood, and thyme essential oils at 23 °C not only inhibited biofilm formation, but it also had bactericidal effect in MH-II and BHI (Figure 7B), while the initial CFU values were reduced from 8 × 10^8^ CFU/mL to 10^2^ in Lime, Peppermint, and Thuja. Less effective were chamomilla, lemon eucalyptus, and petitgrain, which reduced the initial CFU to 10^3^. Citronella, frankincense, lavender oil (Bulgarian and Himalayan), lemon, and Myrtle were 50% effective. When EOs such as ajowan, cinnamon oil (bark, leaf), Citronella, fenugreek, geranium, Lime oil by BP, melissa, palmarosa, rose geranium, and thyme were present in 0.5% concentration in BHI, the CFUs were reduced to 0 at 23 °C. The initial CFU values were reduced from 8 × 10^8^ CFU/mL to 10^4^ by frankincense, Bulgarian lavender oil, patchouli, Vanilla, and Mace.

At an EO concentration of 0.1%, several effective essential oils significantly reduced biofilm formation at 23 °C, with cinnamon (either bark or leaf), geranium, and sandalwood showing 100% efficacy, while palmarosa and thyme reduced the initial CFU to 1 × 10^1^. Ajowan, patchouli, and petitgrain reduced the CFU to 2 × 10^3^, while at a 0.1% concentration, ajowan, geranium, Lime oil as per BP, melissa, palmarosa, rose geranium, and thyme decreased to 0.

Although Cypress essential oil at a 0.5% concentration showed 100% efficiency at 4 °C in the MH-II medium, it was not efficient at higher temperatures. Ajowan, cinnamon (bark and leaves), geranium, palmarosa, petitgrain, and thyme were 100% effective in killing *L. monocytogenes*. Patchouli, Ravensara, rose geranium, and Saffron reduced the initial count (7 × 10^8^) to (2–7) × 10^3^. Pine reduced the number to 1 × 10^4^. In contrast, ajowan, Bay, cinnamon oil (bark, leaf), rose geranium, Lime oil as BP, palmarosa, and thyme reduced the CFU to zero when the experiment was conducted in BHI. Lemon reduced the CFU to 5 × 10^2^. In MH-II, at 4 °C using a concentration of 0.1%, ajowan, cinnamon (bark, leaves), geranium, palmarosa, and thyme showed maximum efficacy, i.e., the CFU was 0. Palmarosa and patchouli reduced the CFU to 1–10^3^, while geranium, pine, Ravensara, rose geranium, and Saffron reduced the CFU to half.

In BHI, still at 4 °C but at a concentration of 0.1%, ajowan, cinnamon (leaf), geranium, thyme showed maximum efficacy, i.e., CFU was 0. Rose geranium reduced the initial CFU to 1 × 10^2^.

At 4 °C, the most effective biofilm-inhibiting essential oils, regardless of their concentration, were ajowan, geranium, Lime oil as per Bp, melissa, Minth, Myrtle, Neroli, palmarosa, patchouli, pine, Ravensara, rose geranium, Saffron, sandalwood, and thyme, with an optical density of less than 0.035. Overall, at all temperatures, the essential oils with the most effective biofilm-inhibiting and CFU reducing properties were ajowan, geranium, Lime oil as per BP, melissa, Minth, Myrtle, Neroli, palmarosa, patchouli, pine, Ravensara, rose geranium, Saffron, sandalwood, and thyme, at concentrations of 0.1 and 0.5%.

Irrespective of the temperature and concentration tested, ajowan, cinnamon, geranium, palmarosa, and thyme were found to be the most effective essential oils, as these EOs completely inhibited CFU and thus biofilm formation (OD: <0.0225). Higher concentrations of essential oils were more effective in all cases. Lemon balm essential oil did not reduce CFUs by 100% in all cases, but inhibited biofilm formation by 100% at all three temperatures tested and at both concentrations (OD: <0.0005), similar to rose geranium (OD: <0.001). It is worth noting that in addition to these EOs, fennel, lemon, eucalyptus, and chamomile monovarietal were also found to be effective in inhibiting biofilm formation, although they did not significantly reduce the CFU.

Regarding the carrier medium for biofilm formation, it was evident that no significant differences between the results of the MH-II and BHI based experiments (*p* = 0.277) could be shown, in contrast to the applied essential oil concentrations and temperatures (*p* < 0.01).

The concentration-dependent effect of EOs in biofilm inhibition is illustrated in the box plot and shows that the more concentrated essential oils were more effective at all temperatures tested. Moreover, the maximum value of biofilms degraded by 0.5% essential oils was lower at all temperatures compared to 0.1% essential oils, so the biofilm was more successfully degraded. Also, the interquartile range is lower, which also supports the idea that the more concentrated essential oil is more effective, and the median is also lower at all temperatures tested using a lower concentration (Figure 9).

### 3.5. Biofilm Eradication from Polystyrene Surfaces

The mature biofilm eradication capacity of the EOs was shown to be temperature and media-dependent.

Higher concentrations of essential oils are generally more effective (B-value: −0.007), and BHI is generally more effective than MH-II (B-value: −0.017). Increasing temperature negatively affects the efficacy of essential oils to destroy complete biofilms (B-value was 0.021 at 23 °C and 0.047 at 37 °C). The efficacy of each essential oil varies and is influenced by the concentration applied, the temperature, and the culture medium (*p* < 0.05).

At 4 °C, the most effective essential oils were palmarosa and geranium, at 23 °C, these were ajowan, geranium, Lime oil by BP, melissa, palmarosa, rose geranium, and thyme, and similar results were obtained at 37 °C. The least effective essential oil at all three temperatures was fenugreek, while at 4 °C it was tea tree, at 23 °C it was Wintergreen, and at 37 °C it was Wintergreen and Jasmine.

At 4 °C, essential oils are more effective in breaking down biofilm in the MH-II medium. An example of this is pine, which was 10 times more effective at breaking down biofilm in MH-II than in BHI. In contrast, the biofilm-disrupting ability of geranium essential oil was not affected by whether BHI or MH-II was used. At 23 °C, the biofilm-disrupting ability of ajowan, geranium, and melissa palmarosa essential oils was not affected by whether they were used in BHI or MH-II.

At 37 °C, the biofilm-disrupting ability of ajowan, geranium, lemon oil according to BP, melissa palmarosa, rose geranium, and thyme essential oils was not affected by whether they were used in BHI or MH-II.

After the biofilm removal test, 96-well plates containing 0.5% essential oils and prepared in MH-II were sampled and the CFU of biofilms was determined and bactericidal effects were shown at all three temperatures.

Ajowan, geranium, Lime oil as per BP, melissa, Orange, palmarosa, rose geranium, and thyme EOs had the most effective antilisterial effects and were also the most effective at degrading the biofilm. Some essential oils were more effective at 4 °C than at lower temperatures. Examples include calamus, chamomilla, Cajeput, Cypress, lavender, Peppermint, and Ravensara. We also found essential oils that were least active at 23 °C: Citronella (Figure 10).

For the 0.1% essential oil biofilm eradication, the narrowest range of values was at 4 °C, while the widest range was at 37 °C. At 4 °C, 50% of the optical values were around 0.04 (630 nm), and this value increased with increasing temperature, indicating that 0.1% essential oils were more effective at lower temperatures. The increased concentration of essential oils was associated with increased efficacy, especially at 4 °C.

A total of 49% of essential oils were most effective at 4 °C, while 15% were most effective at 23 °C, and 36% were most effective at 37 °C. At lower temperatures, the biochemical processes of the bacteria slow down and evaporation is also lower, so the essential oil is more effective at lower temperatures, even at lower concentrations (Figure 11).

Biofilm removal was performed not only in MH-II and BHI but also in CMJ with 57 potential essential oils at 4 and 23 °C. However, the incubation with essential oil was not only performed for 24 h, but also included a 1 h incubation period in the experiment. The figure clearly shows that most essential oils reduced the biofilm at 4 °C, whereas at higher temperatures, most essential oils were able to degrade the biofilm within 1 h. The longer the time interval, the higher was the biofilm eradication efficiency of the 19 essential oils (33%). These essential oils could be potential candidates for preservation, based on their properties. It is noticeable that some essential oils, such as petitgrain and Ravensara, were more effective at 23 °C, while others, such as fennel and Jasmine, were more effective at lower temperatures (Figure 12). The statistical analysis showed a significant difference between the two different temperatures (*p* = 0.002), but no significant difference between the hours and the essential oils used (*p* > 0.05); however, when the hours are examined as a function of temperature, there was a significant difference between the 1 and 24 h results at 4 °C (*p* = 0.045), and a significant difference between the 1 and 24 h results at 23 °C.

The figure (Figure 13) shows that after 1 h of incubation at 4 °C, the box plot values were wider than after 24 h of incubation, so that after 24 h, the essential oils were more effective at 4 °C. At 23 °C, after 1 h of incubation, the values accumulated on a narrower scale when compared with the values of the 1-day experiment. In the latter, the difference between the minimum and maximum values was greater. According to the box plot, the essential oils were most effective at 4 °C after 24 h of incubation (Figure 13).

### 3.6. Biofilm Eradication from SSC

For the essential oils that proved to be more effective, biofilm degradation was demonstrated by SEM. The images show that the tested essential oils are able to degrade the biofilm formed by *L. monocytogenes*. The distorted morphology of the 1860 cells and altered textures of the biofilms formed represented the antibacterial effects when compared to the control (Figure 14). The morphology of untreated bacteria is intact, i.e., bacilli. The intact, thread-like formation connecting each cell is clearly visible (Figure 14A). In contrast, the morphology of treated bacteria is deformed (Figure 14C,G,F,H) and flattened (Figure 14E), while cellular fragments are also clearly visible elsewhere (Figure 14B,D,F).

## 4. Discussion

Efforts to control *L. monocytogenes* in food, and thereby minimize the epidemiological consequences, include a variety of procedures such as chemical, physical, and green technology solutions [52]. In this study, we have focused on the potential application of herbal extracts [13,53,54,55] and the efficacy of the 37 EOs out of the 57 (Figure 2). The EOs investigated was a clear indication that the application of EO-based preservatives targeting *L. monocytogenes* has a raison d’etre, which former studies have also confirmed [13,53,54,55,56,57,58]. However, we also revealed that results of these recent studies should be interpreted with some caution, as the strain, temperature, surface, and certainly concentration of the EO, as well as food type and texture, are crucial determining factors of the antibacterial efficacies of these natural preservatives.

Although former studies demonstrated the efficacy of certain EOs on *L. monocytogenes* [53,54,55,59,60,61,62,63], the effects were only demonstrated on a few isolates. The significant discrepancies in the sensitivities of the 13 strains in our study, however, has revealed that an effective EO may not be necessarily effective for other isolates (Figure 2) and practically may not be applicable to *L. monocytogenes* in general. Despite the noteworthy potential of thyme, cinnamon, and ajowan for food industrial applications, we also revealed that they proved to be effective in a wider range (4–37 °C) of temperature. Thyme [56,57,64,65,66,67,68,69] and cinnamon [70,71,72,73,74] were intensively studied by other authors either in vitro or in different food systems, but to date, there have been few or no studies dedicated to ajowan, rose geranium, and palmarosa, which we have shown to be generally effective against all the 13 *L. monocytogenes* isolates on a wider temperature scale.

In this study, we demonstrated that temperature is an important factor (Figure 2A–D), as it was also formerly demonstrated in the case of cinnamon when it was used in ice cream at a lower concentration (0.1%) and at different temperatures at the lower range (4 °C, 8 °C, 12 °C, 16 °C). The restrictive effect of a low temperature on the bacterial count [75,76] has been demonstrated in other studies performed with different essential oils [75,77], but higher temperatures were not often investigated. From a food preservation point of view, however, this makes sense, as in the case of sous-vide technology, this point is more than relevant. In this process, the meat is heated up to 45–60 °C and then cooled down to 2–8 °C, using both high (for a short term) and low (long term) temperatures. Knowing the fact that *L. monocytogenes* can survive at a high temperature (72 °C) for a short time [78], the most reliable approach in the case of sous-vide technology is to combine it with EOs, such as Salvia, thyme, Rosemary, and Oregano [69,79,80]. This procedure is supported by our findings that the antilisterial effect of several EOs was more pronounced at higher (37 °C) and lower (4 °C) temperatures (Figure 2).

The fluctuation in the antibacterial effect raises some questions. The antibacterial effect is the result of a two-factor game. On one side there is the essential oil, while on the other side there is the target bacterium cell itself. Previous studies have revealed that the stability of essential oils, and thus their antibacterial efficacy, is influenced by temperature, available oxygen, and the presence of light [81,82]. Based on their volatile features, it might be thought that essential oils are more effective at higher temperatures due to a combination of factors related to the chemical composition of the EO and the way temperature affects molecular interactions. Specifically, the increased kinetic energy of molecules at higher temperatures leads to more frequent and energetic collisions, potentially accelerating reactions and enhancing the extraction and release of beneficial compounds, and by that they are able to exert their antibacterial effects most effectively in this range [83]. Our study, however, further refines this picture, at least in the case of *L. monocytogenes,* as several EOs had a firm ability to kill this foodborne pathogen at 37 °C, but also at 4 °C, in such a manner that they lost their antibacterial effect in the intermediate temperature range (14 °C and 23 °C, Figure 2). This observation suggests that not only the physicochemical properties of EOs were detrimental on the effect, but also some features of the bacterial cell [84].

Revealing the underlying molecular events was beyond of the scope of our study, but recent analyses have revealed drastic transcriptomic changes when temperatures around *L. monocytogenes* changed from 37 °C to 25 °C [85]. This temperature shift was shown to affect different metabolic pathways and also chemical tolerances, partly due to the fatty acid composition of the cell membrane [86]. Furthermore, a more drastic change in the structure of the membrane was shown to occur during cold stress, which resulted in a reduced chain length of fatty acids, and increased the concentration of unsaturated fatty acids [87,88]. These changes maintain the fluidity of the membrane at low temperatures and prevent the formation of a gel-like state that could lead to the leakage of the cytoplasmic contents [89]. An increased unsaturated fatty acid content means that the cell membrane becomes more susceptible to hydrophobic small molecules. This explains why hydrophobic compounds of EOs [90] can more easily disrupt the integrity of the bacterial membranes at lower temperatures, ultimately leading to the breakdown of bacterial homeostasis and destruction of metabolism [91]. This more sensitive membrane state of *L. monocytogenes* could explain why the increased EO presence also increases the antibacterial effect, especially at 4 °C (Figure 11).

The lack of temperature-based fluctuations in the antibacterial effect of other EOs, however, might strongly suggest the difference between the mode of actions of EOs and especially their major compounds on the bacterial cell. By losing their antibacterial effects at average temperatures (Figure 2), EOs have antibacterial compounds that cannot penetrate through the cell wall and membrane in the mid temperature range when the composition of the cell wall of *L. monocytogens* changes, in such a way that it becomes impermeable for certain groups of compounds with antibacterial activities.

The altered sensitivity of certain *L. monocytogenes* strains to different EOs can be a crucial aspect of the food industry if the application of essential oil-based preservatives is considered during the cold chain [92]. The demonstration of the temperature fluctuation during the cold chain and the fact that temperature can either reach 18 °C [93], i.e., a temperature where certain EOs lose their antilisterial effect (Figure 2B,C), raises practical issues. Here, we have to mention that the Holliday function of refrigerators are typically around 15 °C, a temperature where several essential oils do not exert their antibacterial effects (Figure 2B).

On the other hand, the sensitivity of ATCC35152 to tarragon EO at lower and higher temperatures (Figure 2A,D), but not at an intermediate temperature (Figure 2B,C), showed an opposite tendency, suggesting another individual mode of action of this EO in relation to this *L. monocytogenes* strain, as certain compounds might penetrate through the membrane in the intermediate temperature range and influence metabolic pathways, such as DNA functions, transcription, translation, and enzyme functions [94,95,96,97,98].

One survival strategy of bacteria to antibacterial agents is their ability to form biofilms on biotic or abiotic surfaces. This self-produced extracellular polymeric matrix provides protection against harsh environmental conditions such as desiccation, nutrient deprivation, or disinfectant treatment [99,100]. Individual differences in biofilm formation between isolates and variable effects of the applied medium and surface qualities (Figure 4 and Figure 5) were supported by previous findings, but to date, no clear associations between biofilm-formation efficiency and genotypes have been found [101]. The recent observations regarding the fact that biofilm formation was positively correlated with the cell surface hydrophobicity and motility, but was independent from planktonic cell growth [29], was supported by our study, as strains 1834 and 1934 showed decreased planktonic growth in BHI (Figure 3C) and also a decreased biofilm-forming capacity (Figure 4(C4)) at 37 °C, but this was fairly temperature and surface-quality-dependent, as at 4 °C, these two strains showed a biofilm-forming capacity that was above average (Figure 4(C1)). This observation confirms that temperature has an influential effect on the biofilm-formation capacity of *L. monocytogenes*. In general, we can state that the pronounced biofilm formation in Mueller Hinton II broth at each temperature (Figure 4(B1,B2,B3,B4)), and the fact that the 13 *L. monocytogenes* isolates grew more intensively in BHI (Figure 3A,D), clearly indicated that MH-II was a better supportive medium for biofilm formation at all four temperatures investigated (Figure 7). Here, the importance of the surface quality has to be emphasized, as for example, the best combination that assured an increased adhesion to the surface of stainless steel was LB at 14 °C (Figure 5). Earlier observations also supported the findings that environmental factors, such as temperature, sugar, salt, pH, nutrients, etc., strongly affect the adhesion and biofilm-formation capacities of Listeria species [102].

As BHI is the classical growth medium for this foodborne pathogen, it has also been used in previous studies for biofilm testing [101,103]. This finding was the reason why the biofilm-formation ability of the 57 essential oils were tested in both mediums and the previously described temperature-dependent effect was tested at three different temperatures.

For biofilm-degradation tests, isolate 1860 was chosen as the ideal candidate, as it had a firm biofilm-forming capacity in all the media tested at all temperatures. This is the reason why strain 1860 was used for a thorough biofilm inhibition and eradication analysis (Figure 8, Figure 10 and Figure 12), similar to strain 13,532 (Appendix A). In addition, these two biofilm-formation tests were partially repeated with strains 1811, 1834, 4a, and 1994, which were representatives for weak and strong biofilm formers (Appendix A). The confirmation of the inhibitory and eliminatory potentials of ajowan, cinnamon, palmarosa, rose geranium, and thyme (Appendix A) showed their general efficacies against *L. monocytogenes*. The efficacy of fennel, lemon eucalyptus, and chamomile to inhibit biofilm formation without showing antibiotic effects, however, strongly suggests the presence of active compounds that specifically affect extracellular matrix formation.

From a practical point of view, antibacterial practices aim not only to inhibit biofilm formation, but to eradicate it and eliminate the microorganism itself from abiotic and food matrices. The quite convincing antibiofilm-forming capacity of ajowan, geranium, Lime, melissa, palmarosa, rose geranium, sandalwood, and thyme at concentrations of 0.1% and 0.5% shown in the cumulative diagrams of biofilm inhibition (Figure 8), mature biofilm degradation (Figure 10), and the balanced antibacterial activities of 1% ajowan and thyme at the investigated temperature range (4–37 °C) studied, emphasize that the pronounced capacity of these latter two essential oils in the control of *L. monocytogenes* in general is outstanding.

## 5. Conclusions

The different sensitivities of the isolates and the temperature-dependent antilisterial effect of the tested EOs need to be taken into account if an EO-based food preservation technology is to be implemented, since several *L. monocytogenes* become resistant to different EOs at intermediate temperatures such as 14 °C and 23 °C. Future studies should be dedicated to revealing whether the membrane structural changes that may be responsible for temperature-dependent sensitivities can be influenced and thereby increase the sensitivity of *L. monocytogenes* either to EOs or to other compounds. From that point of view, the reversed antilisterial effect of tarragon is worth studying to reveal new modes of action.

Based on the antibacterial and antibiofilm testing results of this study, thyme, cinnamon, ajowan, geranium, rose geranium, palmarosa, Lime oil as per BP, Orange, and sandalwood proved to be the most effective candidates for eradication practices.

Fennel, lemon, eucalyptus, and chamomile are good candidates for identifying biofilm-inhibitory compounds and thus directly affecting the expression of this cellular matrix, thereby influencing survival.

Individual differences in the efficacy of EOs and the susceptibility of strains make it reasonable to test EO combinations with overlapping antilisterial effects for future practices.

## Figures and Tables

**Figure 1 foods-14-02097-f001:**
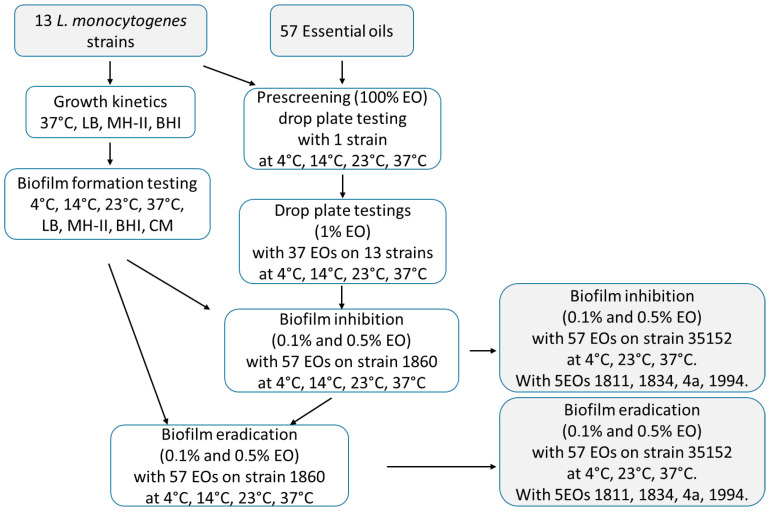
Overview the major steps of the study.

**Figure 2 foods-14-02097-f002:**
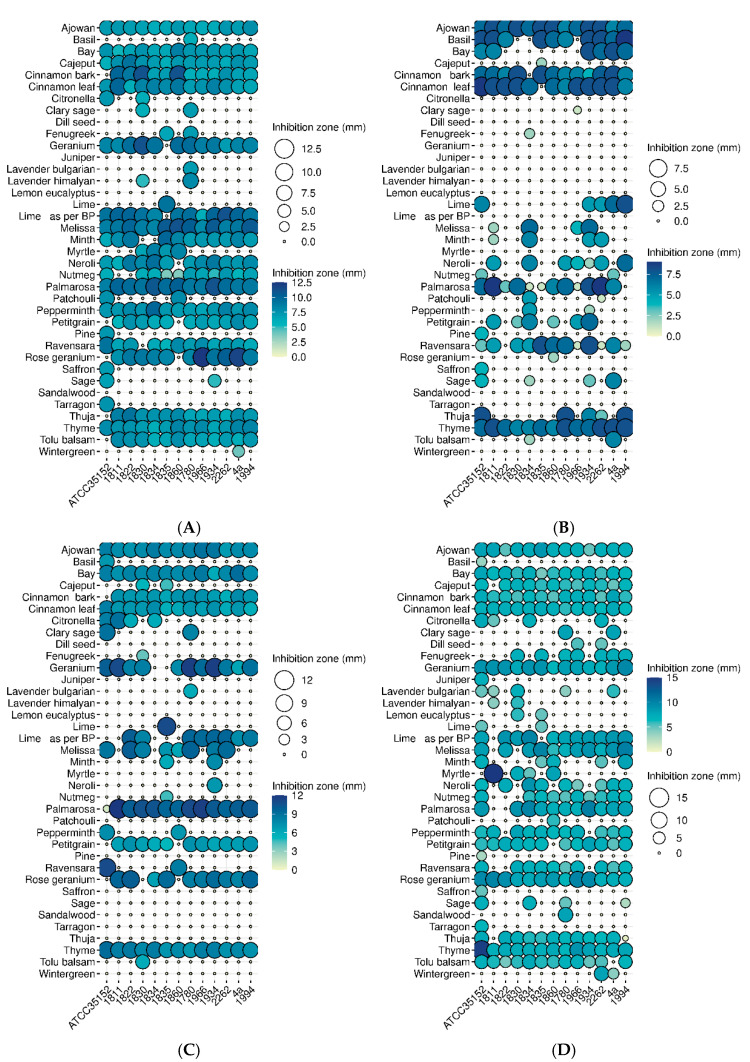
Temperature and strain-dependent antilisterial effects of 37 EOs on 13 different *L. monocytogenes* isolates tested by drop-plate method in 1% concentrations, at (**A**) 4 °C, (**B**) 14°, (**C**) 23 °C, and (**D**) 37 °C.

**Figure 3 foods-14-02097-f003:**
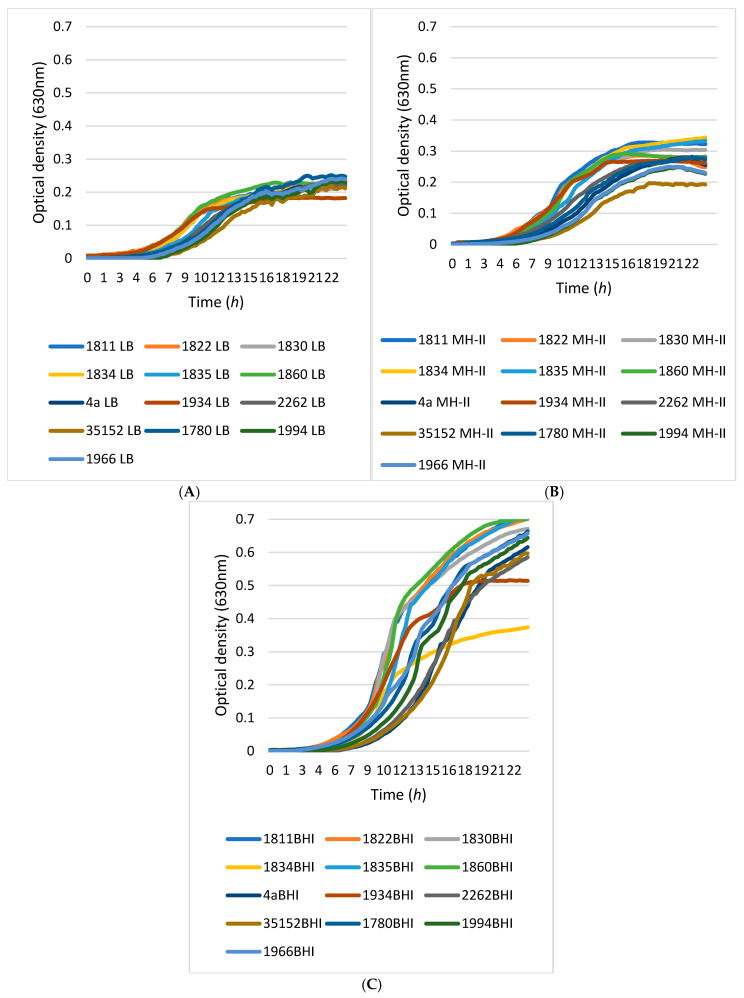
Influence of temperature and carrier medium on the growth kinetics of the 13 *L. monocytogenes* isolates. Experiments were carried out at 23 °C (**A**–**C**) and 37 °C (**D**–**F**) temperatures, while Luria Bertani (**A**,**D**), Mueller Hinton II (**B**,**E**), and Brain Heart Infusion (**C**,**F**) were used as carrier media.

**Figure 4 foods-14-02097-f004:**
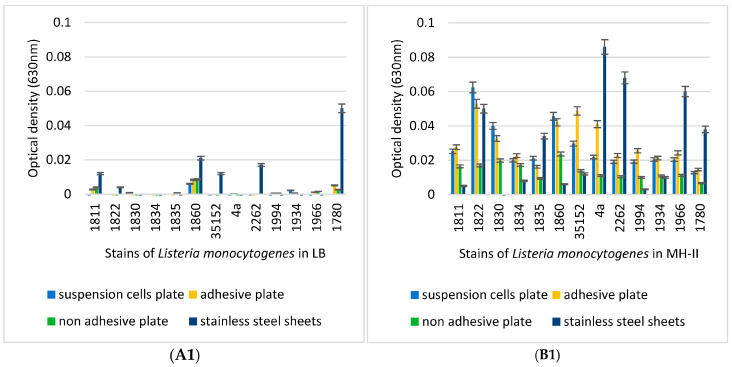
Biofilm-forming abilities of the 13 different *L. monocytogenes* isolates at 4 °C (**A1**–**D1**), 14 °C (**A2**–**D2**), 23 °C (**A3**–**D3**), and 37 °C (**A4**–**D4**) tested in different media such as LB (**A**), MH-II (**B**), BHI (**C**), and CMJ (**D**). (**A1**–**D1**) Biofilm-forming capacity of the 13 different *L. monocytogenes* isolates at 4 °C, in different supportive media such as LB (**A1**), MH-II (**B1**), BHI (**C1**), and CMJ (**D1**). (**A2**–**D2**) Biofilm-forming capacity of the 13 different *L. monocytogenes* isolates at 14 °C, in different supportive media such as LB (**A2**), MH-II (**B2**), BHI (**C2**), and CMJ (**D2**). (**A3**–**D3**) Biofilm-forming capacity of the 13 different *L. monocytogenes* isolates at 23 °C, in different supportive media such as LB (**A3**), MH-II (**B3**), BHI (**C3**), and CMJ (**D3**). (**A4**–**D4**) Biofilm-forming capacity of the 13 different *L. monocytogenes* isolates at 37 °C, in different supportive media such as LB (**A4**), MH-II (**B4**), BHI (**C4**), and CMJ (**D4**).

**Figure 5 foods-14-02097-f005:**
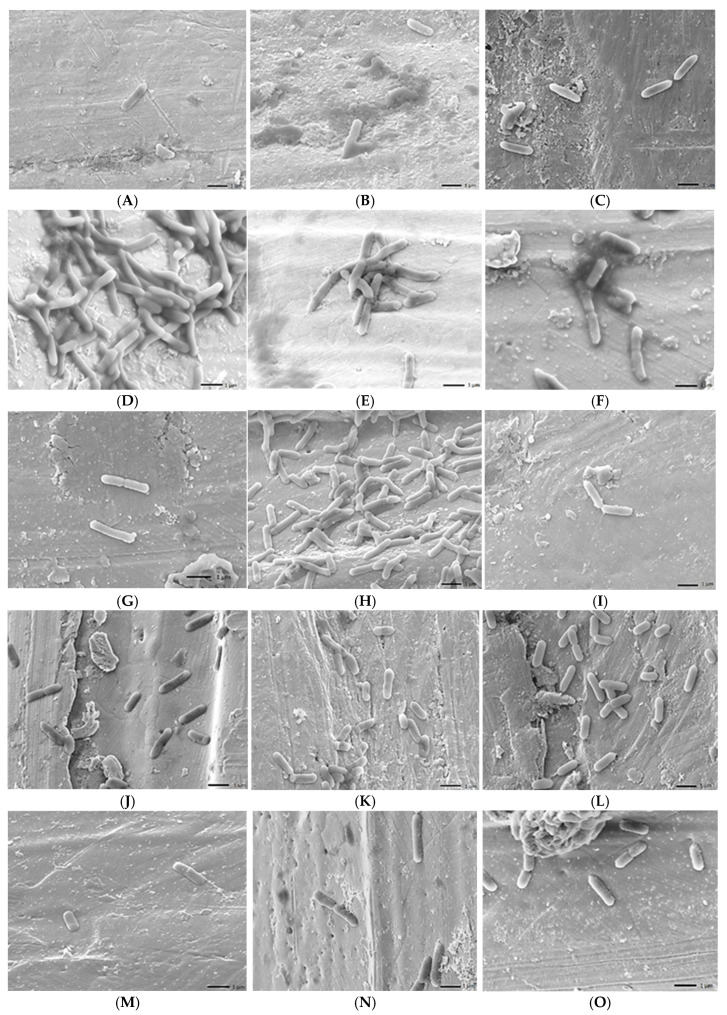
Visualization of the adhesion and biofilm-forming capacities of *Listeria monocytogenes* isolates on the surface of stainless-steel mesh, revealed by scanning electron micrographs at 2000x magnification. (Scale: 1 µm) Isolate 35152: 4 °C: (**A**) (LB), (**B**) (MH-II), and (**C**) (BHI); 14 °C: (**D**) (LB), (**E**) (MH-II), and (**F**) (BHI); 23 °C: (**G**) (LB), (**H**) (MH-II), and (**I**) (BHI); 37 °C: (**J**) (LB), (**K**) (MH-II), and (**L**) (BHI). Isolate 1860: 4 °C: (**M**) (LB), (**N**) (MH-II), and (**O**) (BHI); 14 °C: (**P**) (LB), (**Q**) (MH-II), and (**R**) (BHI); 23 °C: (**S**) (LB), (**T**) (MH-II), and (**U**) (BHI); 37 °C: (**V**) (LB), (**W**) (MH-II), and (**X**) (BHI).

**Figure 6 foods-14-02097-f006:**
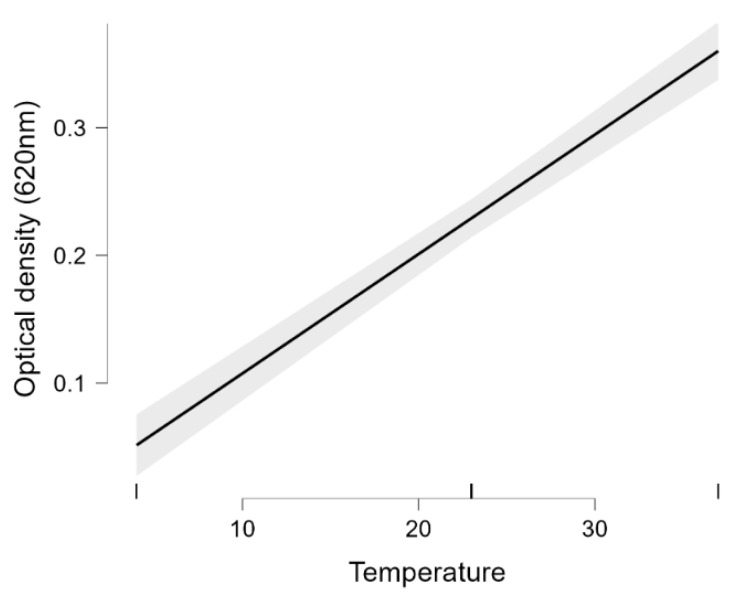
Marginal effect of temperature on biofilm formation. Statistical analysis of the summarised data showed that temperature has a direct supportive effect on biofilm formation in the investigated temperature range (4–37 °C).

**Figure 7 foods-14-02097-f007:**
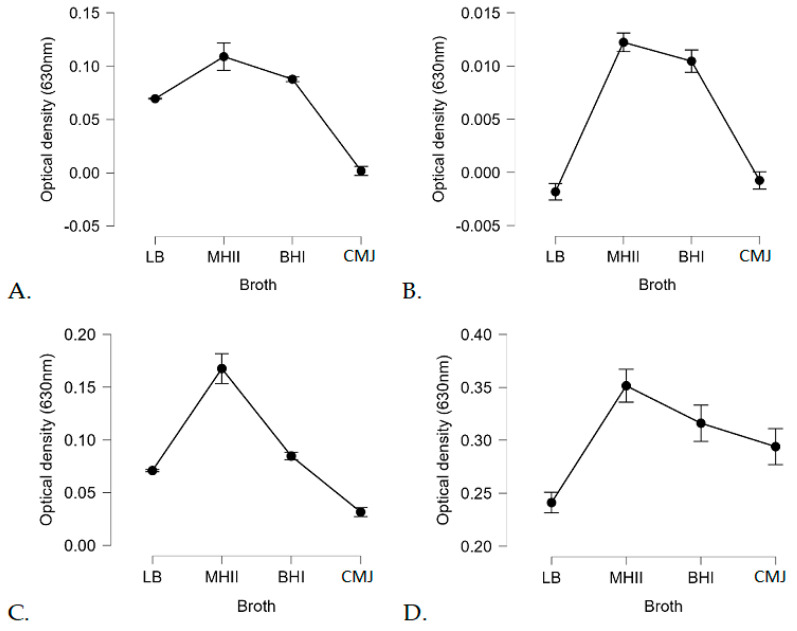
Statistical analysis of the influencing effect of applied media on the biofilm formation at different temperatures, including 4 °C (**A**), 14 °C (**B**), 23 °C (**C**), and 37 °C (**D**), expressed in cumulative OD_620_ values. (Abbreviations: LB: Luria Broth; MH-II: Mueller-Hinton II; BHI: Brain Heart Infusion; CMJ: chicken meat juice).

**Figure 8 foods-14-02097-f008:**
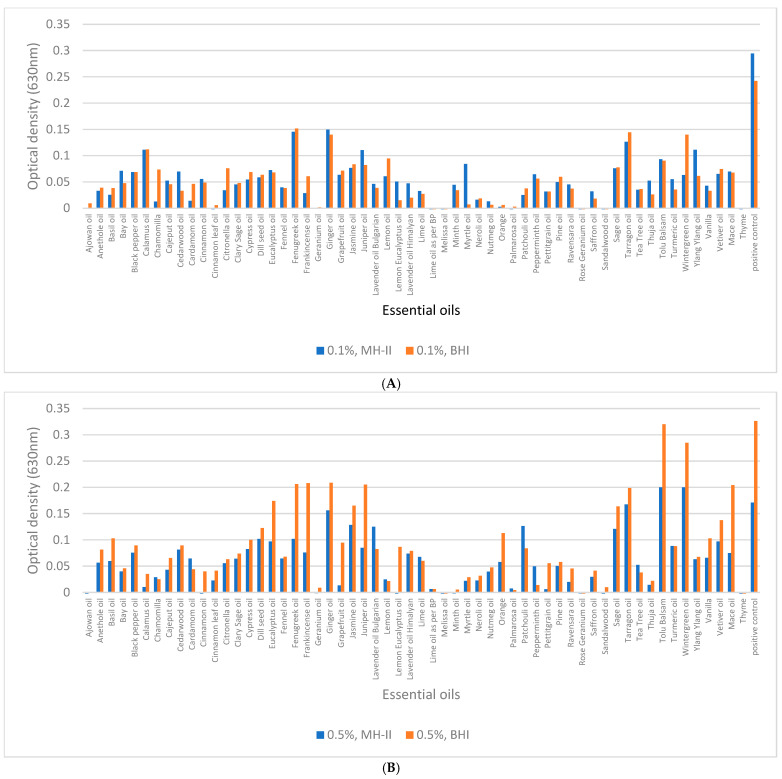
Cumulative diagrams of the biofilm-inhibitory potential of 57 essential oils on the *L. monocytogenes* strain 1860. Bacterium cells were grown in MH-II and BHI in the presence of 0.1% (**A**,**C**,**E**) and 0.5% (**B**,**D**,**F**) EOs, at 37 °C (**A**,**B**), 23 °C (**C**,**D**), and at 4 °C (**E**,F) for 24 h.

**Figure 9 foods-14-02097-f009:**
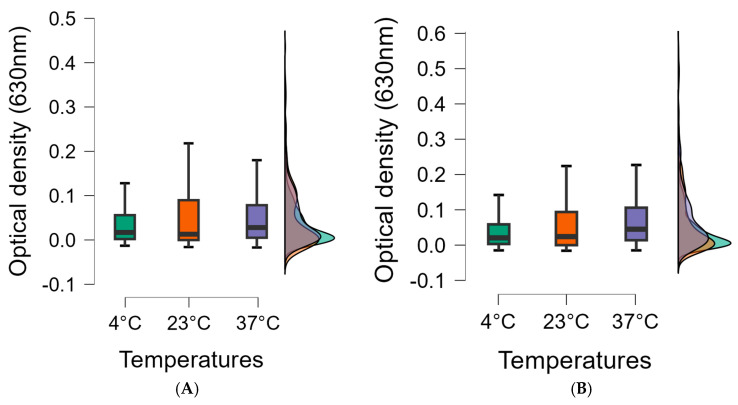
Cumulative optical densities of biofilms inhibited by essential oils as a function of temperature in case of 0.1 (**A**) and 0.5% (**B**) essential oil concentrations. Box plots were used to determine minimum and maximum values, and the section was divided into 3 quartiles: upper, lower quartile, and interquartile range. The middle line of the box represents the median value (50% of our data are in this range), so from this line, half of the data series elements are smaller and half are larger.

**Figure 10 foods-14-02097-f010:**
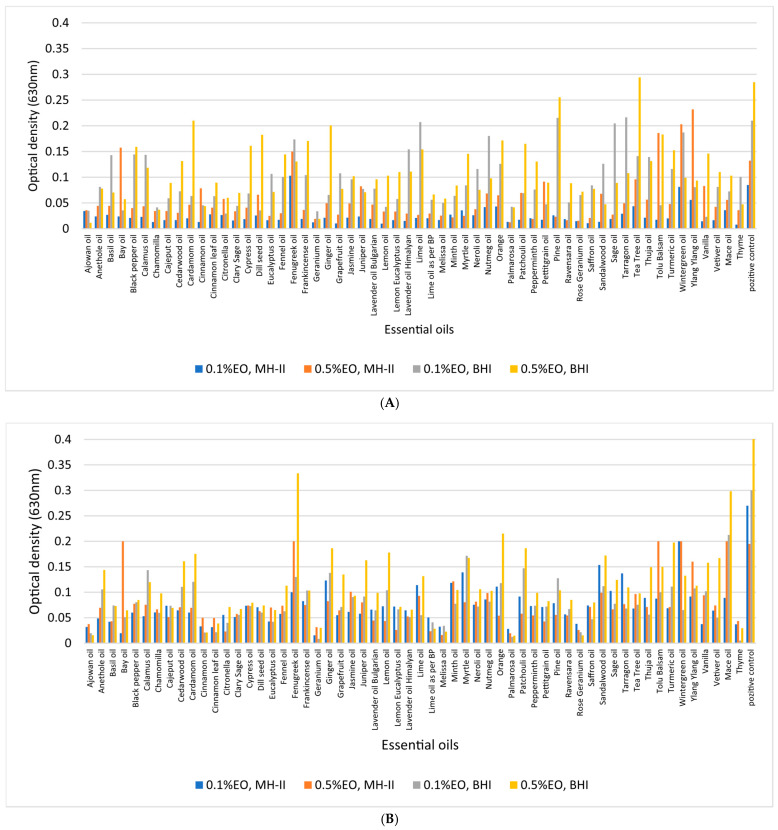
Cumulative results of the biofilm eradication potential of 57 essential oils on the *L. monocytogenes* strain 1860. Bacterium cells were grown in MH-II medium or in BHI at 4 °C (**A**) (for 4 days), 23 °C (**B**) (for 1 day), and 37 °C (**C**) (for 1 day); then, the plates were washed 3 times with PBS. After washing, the plates were filled up with 180 uL MH-II or BHI and 20 uL from the proper concentration of EOs. They were incubated for one more day at 4, 23, or 37 °C.

**Figure 11 foods-14-02097-f011:**
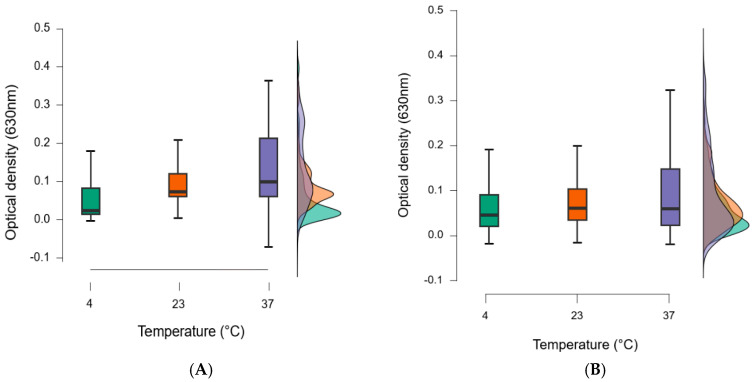
Optical density of biofilms disrupted by essential oil as a function of temperature. Biofilm degradation at 0.1 (**A**) and 0.5% (**B**) essential oil concentration. Box plots are used to determine the minimum and maximum values, and the section is divided into 3 quartiles: upper, lower quartiles, and interquartile range. The middle line in the box represents the median value (50% of our data are in this section), so from here, half of the elements in the data series are smaller and half are larger.

**Figure 12 foods-14-02097-f012:**
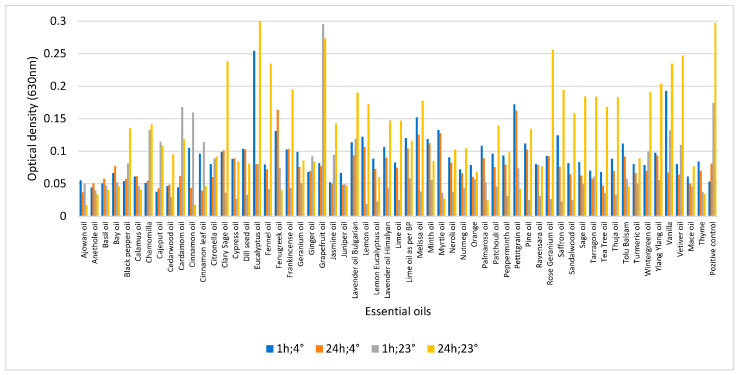
Biofilm removal potential of 57 essential oils on *L. monocytogenes* strain 1860 in CMJ at 4 °C and 23 °C incubated for 1 h and 1 day, respectively. The final concentration of essential oils was 0.5%.

**Figure 13 foods-14-02097-f013:**
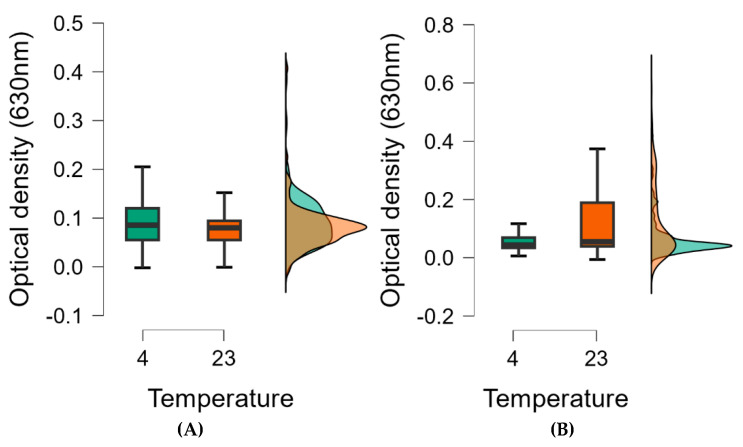
Optical density-based statistical analysis of biofilms disrupted by essential oils as a function of temperature. Biofilm degradation after 1 (**A**) and 24 (**B**) hours of incubation with 0.5% essential oil concentration.

**Figure 14 foods-14-02097-f014:**
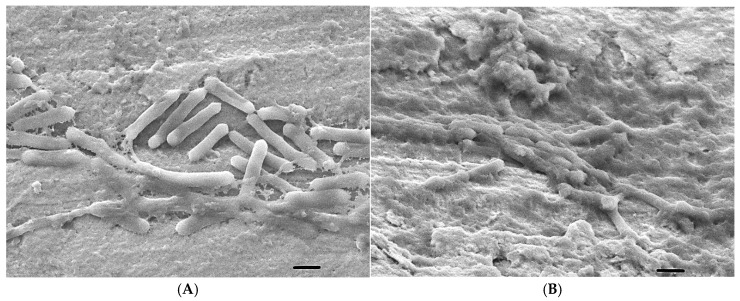
Eradication of adhered *L. monocytogenes* from the surfaces of stainless steel at 4 °C with 0.1%. The formed biofilm was washed 3 times and then essential oils were diluted in MH-II to a final concentration of 0.1%. In the experiment, the positive control was the 1860 strain (**A**), and the essential oils used were basil (**B**), geranium (**C**), Orange (**D**), palmarosa (**E**), Saffron (**F**), tea tree (**G**), and finally thyme (**H**). Scanning electron microscope images at a scale of 10000 × 1 µm, except E, which is 2000 × 10 µm. Black scale at the right bottoms of the images represent 1 µm.

**Table 1 foods-14-02097-t001:** Codes and origins of the *L. monocytogenes* isolates used during the study.

Isolate Abbreviation	Strains of *Listeria monocytogenes*	Origin
1966	NCAIM B01966	NCAIM B01966
1780	NCAIM B01780	NCAIM, LUX
2262	NCAIM B02262	NCAIM, ATCC 19111, OKI133001
1994	NCAIM B01994	NCAIM (Murray, 1926)
1934	NCAIM B 01934	NCAIM, CIP82.110, ATCC 15313, NCTC10357
35152	ATCC 35152	ATCC type strain (Murray 1926)
4a	LmUP_4a	own isolate_2006, raw meat, chicken
1811	LmUP_1811	own isolate_2006, raw meat pork
1822	LmUP_1822	own isolate_2006, raw meat, pork
1830	LmUP_1830	own isolate_2006, goat cheese
1834	LmUP_1834	own isolate_2006, goat cheese
1835	LmUP_1835	own isolate_2006, processed meat
1860	LmUP_1860	own isolate_2006, processed meat

## Data Availability

The original contributions presented in the study are included in the article/Appendix A. Further inquiries can be directed to the corresponding author.

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
