# Peer review of "Effect of Temperature, Surface, and Medium Qualities on the Biofilm Formation of Listeria monocytogenes and Their Influencing Effects on the Antibacterial, Biofilm-Inhibitory, and Biofilm-Degrading Activities of Essential Oils"

_foods, 2025, doi:10.3390/foods14122097_

Round 1
Reviewer 1 Report
Comments and Suggestions for Authors
The manuscript submitted by Seres-Steinbach et al. aimed to evaluate the effects of various essential oils on Listeria monocytogenes biofilms, considering the influence of temperature, surface type, and media properties. The research is highly robust, relevant, and scientifically sound, as it addresses one of the most dangerous foodborne pathogens—L. monocytogenes—whose control remains a challenge in the food industry. The authors investigated a large number of variables, offering important insights into controlling this pathogen. However, the manuscript is difficult to read due to its complexity and dense structure.
The Introduction is well written and requires only minor adjustments. I suggest including data on the economic impact or the number of listeriosis cases to emphasize the pathogen’s significance.
In the Materials and Methods section, the addition of a flowchart outlining the experimental design would be very helpful, as the current format makes it hard for readers to understand the study structure. Furthermore, no references are provided in this section—methodologies must be properly cited to ensure reproducibility and clarity. It is also unclear whether all essential oils were tested at a single concentration. If so, this could have influenced the results regarding media and temperature effects, since each oil may have an optimal concentration.
The Results section is extensive and well organized. However, the Discussion is notably weak and superficial. The authors do not explore the possible mechanisms of action of the most effective essential oils, nor do they discuss the study’s limitations or suggest future research directions. This section needs to be substantially improved.
Other Remarks:
-
Line 23 and elsewhere: Double-check the formatting of all scientific names (italicization, capitalization).
-
Keywords: Replace words already present in the title with others that help contextualize the study and improve indexing.
-
Line 59: Replace on with in (context dependent—please verify).
-
Line 75: Standardize the use of genus abbreviation (e.g., L. for Listeria) consistently throughout the manuscript.
-
Lines 79–82: Cite the “several studies” mentioned.
-
Lines 83–85: Rewrite and expand to better explain the problem being addressed. Avoid extremely short, undeveloped paragraphs.
-
Line 99: Unclear—please verify what is meant here.
-
Section 2.9: This section is currently incomprehensible and poorly presented. Major revision is needed.
-
Throughout the manuscript, including in Section 3.4, use superscript notation correctly for exponential values.
-
Line 667: Essential oils should not be referred to as “herbal extracts”; they are not synonymous.
-
Line 691: Review this sentence—there are grammatical or structural errors.
-
Remove Section 6: It is unnecessary or incorrectly formatted.
-
After the Conclusions, check all subsequent sections. For example, the Supplementary Materials section is included, but no such materials exist. Additionally, the beginning of the Author Contributions section retains placeholder text from the journal template, which must be removed.
Author Response
Reviewer1
The manuscript submitted by Seres-Steinbach et al. aimed to evaluate the effects of various essential oils on Listeria monocytogenes biofilms, considering the influence of temperature, surface type, and media properties. The research is highly robust, relevant, and scientifically sound, as it addresses one of the most dangerous foodborne pathogens—L. monocytogenes—whose control remains a challenge in the food industry. The authors investigated a large number of variables, offering important insights into controlling this pathogen. However, the manuscript is difficult to read due to its complexity and dense structure.
R1: - Thank you for your positive opinion!
The Introduction is well written and requires only minor adjustments. I suggest including data on the economic impact or the number of listeriosis cases to emphasize the pathogen’s significance.
R2: - Data is added.
In the Materials and Methods section, the addition of a flowchart outlining the experimental design would be very helpful, as the current format makes it hard for readers to understand the study structure. Furthermore, no references are provided in this section—methodologies must be properly cited to ensure reproducibility and clarity. It is also unclear whether all essential oils were tested at a single concentration. If so, this could have influenced the results regarding media and temperature effects, since each oil may have an optimal concentration.
R: - Flow chart is added (it is now Figure 1)
- Citing of methods was done.
- In the flow chart the applied concentrations are shown and in the relevant texts (Materials and Methods and Results).
The Results section is extensive and well organized. However, the Discussion is notably weak and superficial. The authors do not explore the possible mechanisms of action of the most effective essential oils, nor do they discuss the study’s limitations or suggest future research directions. This section needs to be substantially improved.
R: - We reviewed Discussion part and made it more didactic focusing on the deeper aspects as You suggested. We improved the english.
Other Remarks:
- Line 23 and elsewhere: Double-check the formatting of all scientific names (italicization, capitalization).
R: scientific names are corrected.
- Keywords: Replace words already present in the title with others that help contextualize the study and improve indexing.
R: With that commence we are a little bit in trouble as we feel that the title is complex so.
- Line 59: Replace onwith in (context dependent—please verify).
R: „on” is in this context is OK.
- Line 75: Standardize the use of genus abbreviation (e.g., for Listeria) consistently throughout the manuscript.
R: done!
- Lines 79–82: Cite the “several studies” mentioned.
R: we thought for the previous studies that were mentioned in the pragrapgh. Now we concretised it.
- Lines 83–85: Rewrite and expand to better explain the problem being addressed. Avoid extremely short, undeveloped paragraphs.
R: the required modifications were done as we entended this paragraph.
- Line 99:
R: it was not relevant. We took it out.
- Section 2.9: This section is currently incomprehensible and poorly presented. Major revision is needed.
R: We just reformulated this section and we hope it is more clear
- Throughout the manuscript, including in Section 3.4, use superscript notation correctly for exponential values.
R: done
- Line 667: Essential oils should not be referred to as “herbal extracts”; they are not synonymous.
R: Thank You for your remark. We corrected it.
- Line 691: Review this sentence—there are grammatical or structural errors.
R: We reformulated the sentence
- Remove Section 6: It is unnecessary or incorrectly formatted.
R: we left this section as it is an interesting point with practical relevance.
- After the Conclusions, check all subsequent sections. For example, the Supplementary Materialssection is included, but no such materials exist.
R: It is now there so we upload this materials.
- Additionally, the beginning of the Author Contributionssection retains placeholder text from the journal template, which must be removed
R: now as we see it is OK.
Reviewer 2 Report
Comments and Suggestions for Authors
Manuscript 3642925
Journal Foods
Title Effect of temperature, surface- and medium qualities on the biofilm formation of Listeria monocytogenes and their influencing effects on the antibacterial, biofilm inhibitory and biofilm degrading activities of essential oils
The manuscript entitled “Effect of temperature, surface- and medium qualities on the biofilm formation of Listeria monocytogenes and their influencing effects on the antibacterial, biofilm inhibitory and biofilm degrading activities of essential oils” describes the effect of temperature, surface, type of medium on the biofilm formation of L. monocytogenes strains. Then, antibiofilm and inhibitory activity of 57 EOs is reported. The topic is not novel, several reports are available in literature focusing on the antibiofilm and antibacterial activity of EOs against L. monocytogenes. Several parts need revision/improvement or explanation. English should be revised throughout the manuscript. Please follow the comments in the file.

Author Response
Reviewer2
Manuscript 3642925 Journal Foods
Title Effect of temperature, surface- and medium qualities on the biofilm formation of Listeria monocytogenes and their influencing effects on the antibacterial, biofilm inhibitory and biofilm degrading activities of essential oils
The manuscript entitled “Effect of temperature, surface- and medium qualities on the biofilm formation of Listeria monocytogenes and their influencing effects on the antibacterial, biofilm inhibitory and biofilm degrading activities of essential oils” describes the effect of temperature, surface, type of medium on the biofilm formation of L. monocytogenes strains. Then, antibiofilm and inhibitory activity of 57 EOs is reported. The topic is not novel, several reports are available in literature focusing on the antibiofilm and antibacterial activity of EOs against L. monocytogenes. Several parts need revision/improvement or explanation. English should be revised throughout the manuscript. Please follow the comments below:
R: Dear reviewer, thank you very much for your opinion.
L18-37 Include the best result with quantitative data
R: we provided the supportive data to the Abstract now.
L39 Remove the numbers 1, 2, 3, 4
R: numbers are removed (Keywords)
L64-65 Add a part on the biofilm-forming ability of L. monocytogenes on different surfaces and its persistence in different environmental conditions
R: we added additional information and aspects concerning to this part (L76-79)
L74-75 Please add references for this statement. The papers doi.org/10.1111/jam.15376 and doi.org/10.3390/antibiotics13040371 are suggested for your analysis and discussion.
R: thank You for your suggestion! We added these relevance papers. (L90)
L86-98 Better describe the experimental workflow in the objective
R: In order to make the study more transparent we inserted a flow chart in Methods part. (please see flow chart)
L103 Why only one reference strain was used and 12 isolates? Please explain
R: To be honest this is not totally correct so, as we used more ATCC strains (Table 1) during the study but the 1934 (ATCC 15313) is the most documented and is also a suggested reference strain in CLSI.
L111 Why CMJ was used? Which is the purpose?
R: The intention behind, to use chicken meat juice was that we wanted to close laboratory tests to practice (slaughterhouse environment), where actually chicken meat can be supportive medium for L. monocytogenes to survive on meat surface or metal surfaces contaminated with meat pieces and liquids (juice). Furthermore, CMJ is much more fatty than the above-mentioned media, so we wanted to investigate the extent to which such a medium affects biofilm formation.
L119-141 Add the chemical composition of all EOs in a supplementary table
R: We do not have an information about that as performing GC-s from 57 EOs was not feasible
L150 Why a sterile blank disc was not used?
R: In our study the simple drop plate method was performed without discs. Certainly before starting to dilute the essential oils in 1% Tween, we tested whether our essential oil-free Tween solution inhibited the 13 L. monocytogenes strains or not. And found that the Tween 20 did not have any influential effect on the bacteria in the applied concentration.
L156-167 Why the growth curve kinetic was followed for 12 isolates? Which is the purpose of this experimental activity? Please explain in the manuscript.
R: This issue was explained in Discussion, and the intention behind was that we wanted to reveal if planktonic growth intensity was in correlation with biofilm intensity or not.
L170-174 Which is the best polystyrene surface for the formation of a biofilm? Please add this information.
R: This features is strain, temperature and medium dependent as it revealed in Figure 3. Forexample at 37°C in the presence of LB, MH-II and CMJ the effects are more or less equaled, but in case of BHI the results between the
L205 How these strains were selected? Explain in the text
R: In materials and methods we mentioned this issue.
L226-228 How the effect of EOs, surface, medium, temperature was evaluated? It is not clear. Rewrite this section
R: we rewrote this section
L229 Check
R: it was a not relevant referring only. We removed it.
L232 Why these 20 EOs were not excluded? In figure 7 all the EOs were considered. Please explain
R: Even though many essential oils did not show an inhibition zone by the drop plate method, this does not mean that they do not have biofilm inhibitory properties, so we also performed biofilm degradation with the essential oils excluded by the drop plate method.
L233 In my opinion, the experimental plan is not balanced. Only these EOs should be considered for the antibacterial and antibiofilm activity against L. monocytogenes. Why authors did not select few EOs?
R: Our purpose with this study was to provide a large scale testing by using different essential oils. The antibacterial effects of certain EOs were formerly revealed by other authors. In frame of this study we identified other candidate EOs, such as Rose geranium, Palmarose with firm antiliteraial effects and others with biofilm inhibiting features.
L237-245 Rewrite this part. It is not clear and correct in English
R: We reformulated this section
Figure 1 Replace this figure with another one showing the heatmap. It is more appropriate as figure R: dear Reviewer thank you for your suggestion, but the statistical software produced this bubble map on that the antibacterial effect can not only seen based on colour change but also the size alterations.
L261-269 Rewrite this part. It is not clear and correct in English
R: we corrected tis section.
L268 Why the temperatures of 4 and 14°C were not included for the growth kinetics? Moreover, why CMJ medium was not considered at this step? Please explain
R: at this stage we only wanted to get a suggestion if media quality has an effect on the growth kinetics.
CMJ was not really suitable for kinetic studies, because, although it was filtered 2x, it becomes more and more opalescent over time, and therefore distorted the measurements very strongly. So practically the background was too high for proper detection.
Figure 2 Replace the time (hours:minutes) with the succession of hours (e.g., 1,6,12,24)
R: done.
L281 Replace polistirol with polystyrene
R: done
L284 Delete at 4°C, 14°C, 23°C and 37°C.
R: it was partially done
L292-297 Please be more specific. Which are the surfaces promoting the biofilm formation?
R: no the information is specified
L281-317 Why a selection of strains/test surface/temperature was not performed? On the basis of the biofilm forming ability, selected cases could be considered for the antibiofilm assays. I do not understand your experimental approach. In figure 7 and 9 only strain 1860 is considered but other strains are relevant for this inhibitory assay at different temperatures and on different surfaces.
R: Thank you very much for your comment. The experiments were also performed on other strains, but for ease of inclusion, only the results of the 1860 isolate have been included in the article. The results on the other isolate were very similar to the 1860 results.
Figure 3/1A-D, Figure 3/2A-D, Figure 3/3A-D, Figure 3/4A-D. Add the statistical analysis and different letters to differentiate mean values
R: This issue is not relevant
Figure 4 should be discussed in the Result section
R: yes it was really omitted, but it is discussed now.
L455-457 Rewrite. It is not clear and correct in English
R: it is rephrased now.
L462-467 Add quantitative data to support the statistical analysis. Which are the parameters most influencing the biofilm formation? Which is the effect of individual parameters? What about the interactive effects between parameters? Revise this part
R: We reformulated this part.
L469 As regards the remaining part of the manuscript, in my opinion, the selection of the most important cases (strain/temperature/surface) to be treated with selected EOs could enhance the clarity and the quality of the manuscript. Please consider to completely revise the experimental plan following this consideration.
R: Thank you very much for your suggestion, but we would stick with this version in the first instance.
L474 Use the same unit for viable cells
R: thank You we synchronise them
L469-486 Use the same unit for viable cells
R: thank You we synchronise them
L469-486 I suppose that the viable cell counts reported in this section are those of planktonic cells within the biofilm matrix. Is it true? Please explain the meaning of viable cell counts reported and the link with the biofilm formation. Now, this part is confusing
R: Thank you very much for your comment. I have revised the manuscript. I hope it is clearer, easier to follow and understand.
L484 reduced CFU to 0 is not appropriate…use a different sentence in this section
R: Done
L485 and by that inhibited biofilm formation also. Revise, it is not correct in English
R: Done
Figure 7/1, 7/2, 7/3 Add the data (OD value?) on the x-axis
R: OD data are on the Y axis.
Figure 7/1, 7/2, 7/3 Use separate columns for MH-II and BHI. Now, it is difficult to follow the results
R: Done
L493-540 Rewrite this part. It is completely not correct in English. It is very hard to follow the sentences. Moreover, some sentences are incomplete. This part has to be checked by a mother-tongue speaker.
R: Done (L562-)
Figure 7/1, 7/2, 7/3 Add the statistical analysis and different letters to differentiate mean values
R: Thank you for your feedback. The problem with this figure that we did not want to make it crowdy with the informations and by that we did not make any thorough statistical analysis in this case.
L553-554 How figure 8 highlights this result? Please explain in the text
R: we mentioned this issue in the text (639-645)
L562-592 Add the Figure 9 to explain the results
R: Done (L682)
Figure 9 Add different columns for the different media.
R: Done
L602-603 Why? Please explain in the manuscript
R: explained in discussion where membrane composition is discussed. (L832-849)
L605-608 How many EOs were more effective at 4°C than high temperature? Add this info
R: We did not want to specify this number, because there are some differences in the patterns among individual EOs. So individual EOs has to be checked for this info if are interested by the reader.
L619-621 Rewrite this part. It is not correct in English.
R: we reformulated this part.
L624-628 Better explain this part.
R: we tried to explain this issue at the relevant part
Figure 11 Add different columns for the different temperatures and time of incubation
R: Done
L646-650 Expand this part. Which are the morphological changes detected?
R: Done. Now it is from 741-748.
L691 Complete this part: like ….
R: Discussion is completely rewritten now, so this became not relevant.
L697-700 Why certain EOs are more effective at high temperature? Please discuss this result using relevant references. Add a possible explanation and support this statement with the help of literature
R: we explained this issue. (L832-849)
L723-731 Are these results potentially linked with the stress-response of L. monocytogenes strains to low temperature conditions? Discuss this aspect with relevant references
R: This issue is discussed. Where it was referred that cold stress could evoke the membrane changes that influence sensitivity to EOs.
L747-767 This is the Discussion Section. These results are already reported but, here, they are not discussed in depth. Please discuss these results
R: We reformulated this section as we did not want to go in detail why BHI is a better medium. (L912-917)
Conclusion section Please add the influence of surface on the antilisterial and antibiofilm activity of EOs
R: these aspects are added.

Round 2
Reviewer 2 Report
Comments and Suggestions for Authors
Authors revised the original version according to reviewer's comments. Minor changes are suggested below:
Original comment: L281-317 Why a selection of strains/test surface/temperature was not performed? On the basis of the biofilm forming ability, selected cases could be considered for the antibiofilm assays. I do not understand your experimental approach. In figure 7 and 9 only strain 1860 is considered but other strains are relevant for this inhibitory assay at different temperatures and on different surfaces.
R: Thank you very much for your comment. The experiments were also performed on other strains, but for ease of inclusion, only the results of the 1860 isolate have been included in the article. The results on the other isolate were very similar to the 1860 results.
Please include the results of other strains in supplementary material
Author Response
Review process 2nd stage
Reviewer2
Original comment: L281-317 Why a selection of strains/test surface/temperature was not performed? On the basis of the biofilm forming ability, selected cases could be considered for the antibiofilm assays. I do not understand your experimental approach. In figure 7 and 9 only strain 1860 is considered but other strains are relevant for this inhibitory assay at different temperatures and on different surfaces.
R: Thank you very much for your comment. The experiments were also performed on other strains, but for ease of inclusion, only the results of the 1860 isolate have been included in the article. The results on the other isolate were very similar to the 1860 results.
Please include the results of other strains in supplementary material
R: Biofilm formation inhibitory and mature biofilm degradation capacities of the 57 EOs at different concentrations were summarised in figure 7 and 9 respectively. Only strain 1860 was chosen for the detailed experiments as it showed a generally firm biofilm forming capacity, similar to strain 35152. With this latter strain the experiments were repeated, but aftre it we chosen out two weak (1811 and 1834) and two strong biofilm formers (4a and 1994) and tests were carried out with them, but only with those essential oils that showed the best performance for eradication like Ajowain, Cinnamon, Palmarosa, Pose geranium and Thyme (SFigure3). So actually that could be disturbing in our answer tests were carried out with the best perfoming essential oils and altogether 4 selected strains, 2 with weak, and 2 with strong biofilm forming capacities.
These results we summarised in 3 Supplementary Figures that w attached now and relevant references were inserted in the text also (L925-931).
And this information was also added to Figure 1.